# Biodiversity of Coleoptera (Insecta) in the Middle and Lower Volga Regions (Russia)

Leonid V. Egorov [1], Sergei K. Alekseev [2], Alexander B. Ruchin [3], Aleksey S. Sazhnev [4], Oleg N. Artaev [4,*], Mikhail N. Esin [3], Evgeniy A. Lobachev [5], Sergei V. Lukiyanov [5], Anatoliy V. Semenov [3], Yulia A. Lukyanova [6], Nikolai V. Shulaev [7] and Kirill V. Litvinov [8]

[1] Prisursky State Nature Reserve, 428034 Cheboksary, Russia
[2] Ecological Club "Stenus", 248000 Kaluga, Russia
[3] Joint Directorate of the Mordovia State Nature Reserve and National Park Smolny, 430005 Saransk, Russia
[4] Papanin Institute for Biology of Inland Waters, Russian Academy of Sciences, 152742 Borok, Russia
[5] Faculty of Biotechnology and Biology, National Research Mordovia State University, 430005 Saransk, Russia
[6] National Park Nizhnyaya Kama, 423600 Elabuga, Russia
[7] Institute of Fundamental Medicine and Biology, Kazan Federal University, 420008 Kazan, Russia
[8] Astrakhan Biosphere Nature Reserve, 414021 Astrakhan, Russia
* Correspondence: artaev@gmail.com; Tel.: +7-83445-296-35

**Abstract:** (1) Background: The conservation of entomofauna in individual macroregions requires efforts to study the distribution and abundance of insects. For this purpose, databases are created that enumerate this information. Such databases, with the processing of significant factual material, make it possible to objectively assess the status of a species and, if necessary, take measures for its protection. The aim of the paper is to describe the modern Coleoptera fauna in nine regions of Russia on the basis of a recently published dataset. (2) Methods: We conducted our own studies in 1994, 1996, 1998–2003 and 2005–2022. The dataset also includes data from museum specimens from other years. We used a variety of methods, such as sifting through litter, searching under the bark of trees and stumps, trapping by light, soil traps, beer traps, window traps, etc. For each observation, the coordinates of the find, the number of individuals observed and the date were recorded. (3) Results: The dataset contains data on 1469 species and subspecies of Coleoptera from 85 families found in the Volga Region. In total, there are 31,433 samples and 9072 occurrences in the dataset. (4) Conclusions: The largest families in terms of species diversity are Curculionidae (202 species), Carabidae (145 species) and Chrysomelidae (142 species). There are 54 species of Coleoptera with a northern range boundary in the macroregion, two species with a southern range boundary and one species with an eastern range boundary. Twenty-one invasive Coleoptera species have been recorded in the macroregion.

**Keywords:** species diversity; beetles; species conservation; dataset; European Russia

## 1. Summary

Many insect species are at risk of extinction or significant decline due to anthropogenic factors [1]. There are numerous well-documented examples of insect population declines and extinctions in different regions of the world due to a variety of anthropogenic causes. Such population declines on a local scale threaten to fragment habitats [2–8]. Didham et al. [1] identified seven key challenges in drawing reliable conclusions about insect population declines. However, one important factor to consider is the lack of distribution and population data for many insect species from different macroregions. Conservation of a particular endangered insect species must be based on accurate knowledge of abundance

and distribution, the causes and extent of decline, the identification of threats and monitoring programmes to assess conservation status [9–12]. Unfortunately, for many insects of conservation concern, distribution data are often lacking due to poor knowledge of their ecology and little study of range boundaries and particular macro-regions [2,13–17]. Moreover, long-term fluctuations in insect abundance are complex and multifaceted, which indicates the need for multi-year studies of the same macroregions. It is therefore necessary to be very careful when assessing the evidence on population trends and identifying the drivers of these trends [1,5,18,19].

With its vast territory and high diversity of natural habitats, Russia faces enormous challenges in its efforts to conserve its entomofauna [20]. The most obvious problem is the lack of basic information on the status and abundance trends of all insect species [20,21], except for a few well-studied groups [22,23]. There is thus an ongoing need to document and understand changes in insect abundance and distribution data [24,25]. To this end, database platforms have been created that all users can access [26–28]. These platforms provide the largest collections of species occurrence records [29,30]. Despite the considerable progress that has been made in the field of open data in ecology over the last decade, there is still room for improvement. However, the accumulation of insect data should and is taking place on a significant scale [29,31].

The aim of this paper is to describe the modern Coleoptera fauna in a large macroregion which covers an area of more than 539,000 km$^2$ and extends over 1000 km from north to south on the basis of a recently published dataset [32].

## 2. Data Description

### 2.1. Dataset Name

In the dataset, each observation includes basic information: date of observation, coordinates (latitude/longitude), observer name, identifier name and publications (if available). The coordinates were determined in situ using a GPS device or after surveys using Google Maps (Table 1).

**Table 1.** Description of the data in the dataset.

| Column Label | Column Description |
| --- | --- |
| occurrenceID | An identifier for the occurrence (as opposed to a particular digital record of the occurrence) |
| basisOfRecord | The specific nature of the data record: HumanObservation |
| scientificName | The full scientific name, including the genus name and the lowest level of taxonomic rank with the authority |
| kingdom | The full scientific name of the kingdom in which the taxon is classified |
| decimalLatitude | The geographic latitude of location in decimal degree |
| decimalLongitude | The geographic longitude of location in decimal degrees |
| geodeticDatum | The ellipsoid, geodetic datum or spatial reference system (SRS) upon which the geographic coordinates given in decimalLatitude and decimalLongitude are based |
| country | The name of the country in which the location is found |
| countryCode | The standard code for the country in which the location is found |
| individualCount | The number of individuals present at the time of the occurrence |
| eventDate | The date when material from the trap was collected or the range of dates during which the trap collected material |
| year | The integer day of the month on which the event occurred |
| month | The ordinal month in which the event occurred |
| day | The integer day of the month on which the event occurred |
| recordedBy | A person, group or organization responsible for recording the original occurrence |
| identifiedBy | A list of names of the people who assigned the taxon to the subject |
| bibliographicCitation | A related resource that is referenced or pointed to by the described resource |

### 2.2. Figures, Tables and Schemes

The dataset contains data on 1469 species and subspecies of Coleoptera from 85 families found in nine regions of European Russia (Nizhny Novgorod Region, Saratov Region, Ulyanovsk Region, Samara Region, Volgograd Region, Astrakhan Region, Chuvash Republic, Republic of Tatarstan, Republic of Mari El) and documented simultaneously with coordinates (Table 2). The total number of specimens in the dataset is 9072; the number of specimens represented is 31,433. Curculionidae (202), Carabidae (145) and Chrysomelidae (142) are the largest families in terms of species diversity.

**Table 2.** Species diversity of Coleoptera families from the dataset.

| Family | Number of Species | Number of Individuals |
|---|---|---|
| Gyrinidae | 5 | 49 |
| Haliplidae | 12 | 65 |
| Noteridae | 2 | 18 |
| Dytiscidae | 76 | 613 |
| Carabidae | 145 | 1441 |
| Scirtidae | 11 | 336 |
| Eucinetidae | 1 | 4 |
| Dascillidae | 1 | 1 |
| Byrrhidae | 3 | 11 |
| Buprestidae | 18 | 139 |
| Dryopidae | 2 | 4 |
| Elmidae | 3 | 23 |
| Limnichidae | 1 | 1 |
| Heteroceridae | 12 | 621 |
| Throscidae | 2 | 2 |
| Eucnemidae | 10 | 31 |
| Lycidae | 5 | 69 |
| Cantharidae | 20 | 402 |
| Elateridae | 50 | 869 |
| Drilidae | 1 | 3 |
| Lampyridae | 1 | 37 |
| Histeridae | 25 | 143 |
| Georissidae | 1 | 1 |
| Helophoridae | 4 | 12 |
| Hydrochidae | 4 | 8 |
| Hydrophilidae | 49 | 909 |
| Ptiliidae | 2 | 2 |
| Hydraenidae | 6 | 20 |
| Leiodidae | 16 | 58 |
| Staphylinidae | 74 | 719 |
| Trogidae | 1 | 2 |
| Lucanidae | 5 | 214 |
| Bolboceratidae | 1 | 5 |
| Geotrupidae | 3 | 38 |
| Scarabaeidae | 67 | 3963 |
| Dermestidae | 16 | 345 |
| Ptinidae | 14 | 234 |
| Byturidae | 2 | 30 |
| Biphyllidae | 1 | 3 |
| Cleridae | 6 | 57 |
| Trogossitidae | 4 | 11 |
| Melyridae | 9 | 282 |
| Lymexylidae | 1 | 1 |
| Mordellidae | 6 | 165 |
| Ripiphoridae | 1 | 1 |
| Scraptiidae | 5 | 169 |

**Table 2.** *Cont.*

| Family | Number of Species | Number of Individuals |
|---|---|---|
| Oedemeridae | 8 | 184 |
| Mycteridae | 1 | 1 |
| Aderidae | 3 | 24 |
| Boridae | 1 | 2 |
| Pythidae | 1 | 1 |
| Salpingidae | 4 | 8 |
| Pyrochroidae | 3 | 41 |
| Meloidae | 6 | 13 |
| Anthicidae | 7 | 19 |
| Melandryidae | 15 | 83 |
| Zopheridae | 3 | 5 |
| Ciidae | 21 | 75 |
| Tetratomidae | 1 | 2 |
| Mycetophagidae | 10 | 139 |
| Tenebrionidae | 33 | 744 |
| Bothrideridae | 1 | 4 |
| Cerylonidae | 5 | 65 |
| Latridiidae | 18 | 839 |
| Corylophidae | 3 | 16 |
| Endomychidae | 4 | 27 |
| Coccinellidae | 43 | 674 |
| Erotylidae | 11 | 179 |
| Sphindidae | 3 | 69 |
| Monotomidae | 8 | 85 |
| Kateretidae | 5 | 12 |
| Nitidulidae | 25 | 7988 |
| Cryptophagidae | 7 | 63 |
| Cucujidae | 3 | 38 |
| Silvanidae | 6 | 18 |
| Phalacridae | 6 | 12 |
| Laemophloeidae | 5 | 9 |
| Orsodacnidae | 1 | 25 |
| Cerambycidae | 84 | 3359 |
| Chrysomelidae | 142 | 1194 |
| Cimberididae | 1 | 2 |
| Anthribidae | 6 | 18 |
| Attelabidae | 8 | 83 |
| Brentidae | 51 | 683 |
| Curculionidae | 202 | 2499 |
| **Total** | **1469** | **31,433** |

The following families were represented by only one species: Eucinetidae, Dascillidae, Drilidae, Lampyridae, Georissidae, Trogidae, Bolboceratidae, Biphyllidae, Lymexylidae, Boridae, Pythidae, Tetratomidae, Bothrideridae, Limnichidae, Mycteridae, Ripiphoridae, Cimberididae.

There are borders of distribution for many Coleoptera species in the macroregion. Due to the considerable latitudinal extent of the studied macroregion, a large number of the species have northern borders of distribution: *Notoxus binotatus* (Gebler, 1829), *Notoxus simulans* Heberdey, 1935, *Notoxus trifasciatus* Rossi, 1792, *Anthaxia semicuprea* Küster, 1851, *Acinopus laevigatus* Ménétriés, 1832, *Acinopus picipes* (G.-A. Olivier, 1795), *Cephalota elegans* Fischer von Waldheim, 1823, *Cicindela campestris pontica* Fischer von Waldheim, 1828, *Taphoxenus goliath* (Faldermann, 1836), *Cymindis miliaris* (Fabricius, 1801), *Harpalus hospes hospes* Sturm, 1818, *Agapanthia cardui* (Linnaeus, 1767), *Agapanthia violacea* (Fabricius, 1775), *Cerambyx scopolii* Fuessly, 1775, *Ropalopus ungaricus insubricus* (Germar, 1823), *Stenocorus quercus* (Gotz, 1783), *Vadonia bipunctata* (Fabricius, 1781), *Bruchus sibiricus* Germar, 1823, *Labiaticola sibiricus* (Faust, 1890), *Tychius astragali* Becker, 1862, *Tychius tridentinus* Penecke,

1922, *Hydaticus grammicus* Sturm, 1834, *Hydrovatus cuspidatus* (Kunze, 1818), *Hygrotus corpulentus* (Schaum, 1864), *Hygrotus flaviventris* (Motschulsky, 1859), *Hygrotus saginatus* (Schaum, 1857), *Porhydrus obliquesignatus* (Bielz, 1852), *Aeolosomus rossii* (Germar, 1844), *Agriotes ustulatus* (Schaller, 1783), *Augyles flavidus* (Rossi, 1794), *Augyles intermedius* (Kiesenwetter, 1843), *Augyles maritimus* (Guerin-Meneville, 1844), *Berosus bispina* Reiche & Saulcy, 1856, *Berosus frontifoveatus* Kuwert, 1888, *Heterocerus heydeni* Kuwert, 1890, *Hydrochara dichroma* (Fairmaire, 1892), *Hydrochara flavipes* (Steven, 1808), *Hydrophilus piceus* (Linnaeus, 1758), *Limnoxenus niger* (Gmelin, 1790), *Pyrochroa serraticornis* (Scopoli, 1763), *Geotrupes spiniger* (Marscham, 1802), *Agoliinus isajevi* (Kabakov, 1994), *Caccobius histeroides* (Menetries, 1832), *Ceratophyus polyceros* (Pallas, 1771), *Cheironitis pamphilus* (Ménétriés, 1849), *Polyphylla fullo* (Linnaeus, 1758), *Sisyphus schaefferi* (Linnaeus, 1758), *Tropinota hirta* (Poda von Neuhaus, 1761), *Protaetia caucasica* (Kolenati, 1846), *Aspidiphorus lareyniei* Jacquelin du Val, 1859, *Cteniopus sulphureus* (Linnaeus, 1758), *Gonocephalum granulatum pusillum* (Fabricius, 1792), *Helops caeruleus stevenii* Krynicki, 1834, *Synchita separanda* (Reitter, 1882). At the southern border of the range, there are two species: *Agabus affinis* (Paykull, 1798) and *Agabus biguttatus* (G.-A. Olivier, 1795). *Valgus hemipterus* (Linnaeus, 1758) and *Notoxus binotatus* (Gebler, 1829) have eastern and western limits, respectively.

The findings on invasive Coleoptera species are interesting. The dataset contains information on invasive species that are common in the regions studied: *Anthrenus picturatus* (Solsky, 1876), *Agrilus planipennis* Fairmaire, 1888, *Atomaria lewisi* Reitter, 1877, *Carpophilus hemipterus* (Linnaeus, 1758), *Cercyon laminatus* Sharp, 1873, *Cryptophagus punctipennis* C.N.F. Brisout de Barneville, 1863, *Cryptopleurum subtile* (Sharp, 1884), *Dermestes lardarius* Linnaeus, 1758, *Glischrochilus quadrisignatus* (Say, 1835), *Harmonia axyridis* (Pallas, 1773), *Lasioderma serricorne* (Fabricius, 1792), *Leptinotarsa decemlineata* (Say, 1824), *Necrobia violacea* (Linnaeus, 1758), *Omonadus floralis* (Linnaeus, 1758), *Omosita discoidea* (Fabricius, 1775), *Psylliodes hyoscyami* (Linnaeus, 1758), *Tenebrio molitor* Linnaeus, 1758, *Trichoferus campestris* (Faldermann, 1835), *Trogoderma glabrum* (Herbst, 1783), *Trogoderma versicolor* (Creutzer, 1799), *Typhaea stercorea* (Linnaeus, 1758).

*Agrilus planipennis* is considered the most dangerous pest of ash in the world [33]. Before 2015, infestation of this species affected central regions of Russia [34]; in 2018–2020, the pest began to occur much further south and west [33]. In 2018 and 2019, it was found in the Volgograd region [35]. Apparently, it will appear in the Saratov, Samara and Ulyanovsk regions in the near future.

The current spread of *Harmonia axyridis* out of its native range is probably due to unintentional introduction by adults via fruit or movement in transport [34]. The species often first appears exclusively in anthropogenic habitats (cities, villages and near roads) but is later observed in natural habitats as well [36]. In 2018, it started to occur in the Volgograd Region, and in 2019 in the Astrakhan, Saratov, Samara and Ulyanovsk Regions and in the Republic of Tatarstan and Chuvash Republic [36].

*Trichoferus campestris* in the studied macroregion was first found in the Astrakhan Region in 1988 [34]. At present, it is widely dispersed everywhere. It is found in a wide variety of habitats, both natural and anthropogenic. Adults do not feed [34]; however, they have been repeatedly observed in traps with fermented bait [37].

## 3. Methods

The macroregion selected for this research includes 9 regions of Russia (Nizhny Novgorod Region, Saratov Region, Ulyanovsk Region, Samara Region, Volgograd Region, Astrakhan Region, Chuvash Republic, Republic of Tatarstan, Republic of Mari El). The total area of these regions is 539,320 km$^2$. These regions are united by the Volga River, which runs through their territories (Figure 1). Geographically, the study sites are located on the East European Plain. Most of the regions in which modern studies have been carried out lie within the Volga Upland and partly within the Ergeni Upland, the Kalach Upland, the Don Ridge, Vyatsky Uval and several lowlands (Mari, Pre-Caspian, Oka-Don and Khoper-Buzuluk).

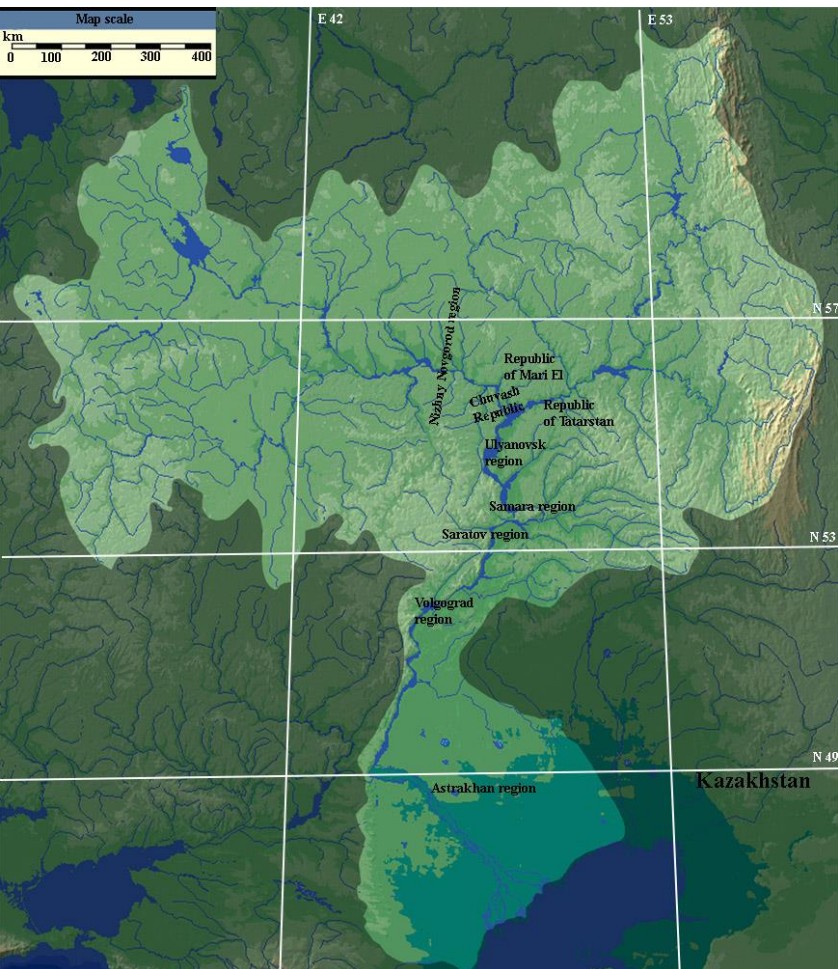

**Figure 1.** Map of the studied regions.

The macroregion stretches meridionally from north to south for more than 1300 km, so it lies within several climatic zones (taiga, mixed forests, forest-steppes and steppes). In some places, these zones replace each other; in others, they are interspersed with intrazones. Climatic variation within the macroregion is quite pronounced. The climate is moderately continental, with distinct seasons [38]. There is the following regularity: when moving from north-west to south-east, the continental climate increases, as the eastern and south-eastern influences strengthen and the western and north-western ones weaken. However, in some areas this pattern is disturbed because of local conditions, such as elevated topography and vegetation. We also note the frequent occurrence of climatic anomalies, such as strong winds in some areas.

In the northern half of the macroregion, the relative humidity is noticeably higher than in the southern half. Thus, the average annual relative humidity is 78% in Ulyanovsk and 70% in Saratov. This indicates an increase in the aridity of the climate from north to south. In winter, the highest relative humidity is observed throughout the territory, while in summer it is particularly low, with relative humidity falling particularly low in the southernmost areas [38].

The Volga–Don watershed runs through the studied macroregion. In the northern part, the watershed line extends far from the Volga River and is west of the Sura River. In the south, it approaches the Volga almost closely, so that the Volga and Don slopes are extremely unevenly developed. The major rivers that originate or flow on the Volga Upland are the Sura River, the Sviyaga River, the Khoper River, the Medveditsa River, the Ilovlya River and a number of smaller rivers.

The material studied in this paper consisted of Coleoptera collected mainly in 1994, 1996, 1998–2003 and 2005–2022. The dataset used also includes data from museum specimens collected in other years and information from some publications of recent years. Information in the dataset that came from the publications of other authors was checked and entered with exact coordinates. Collections were made using a variety of means, such as sifting through litter, searching under tree bark and stumps, collecting in decaying substrates, splashing and trampling on pond banks, catching by light, in soil traps, beer traps, window traps, etc. [39,40]. The surveys were carried out in all districts of the region and over 600 geographical locations were surveyed. The specimens are held in the collections of the Mordovia State Nature Reserve (Pushta, Republic of Mordovia), the Zoological Institute (St. Petersburg), the Zoological Museum of Moscow State University (Moscow), the State Nature Reserve "Prisursky" (Cheboksary, Chuvash Republic), as well as in personal collections of the authors.

The classification of taxa into families was made using modern data [41,42]. The species lists were checked according to the Catalogue of Palaearctic Coleoptera [43–51] and other publications [52,53]. The years of description for some beetle species are given according to Bousquet [54]. The list of invasive species is given according to the reference book [34].

**Author Contributions:** Conceptualization, L.V.E.; methodology, A.B.R. and O.N.A.; software, O.N.A. and A.V.S.; validation, L.V.E. and A.B.R.; formal analysis, L.V.E. and S.K.A.; investigation, A.S.S., M.N.E., E.A.L., S.V.L., N.V.S., Y.A.L. and K.V.L.; resources, A.B.R., A.V.S., N.V.S., Y.A.L., A.S.S. and M.N.E.; data curation, O.N.A. and A.B.R.; writing—original draft preparation, L.V.E. and A.B.R.; writing—review and editing, L.V.E.; visualization, L.V.E.; supervision, L.V.E.; project administration, O.N.A.; funding acquisition, A.B.R. All authors have read and agreed to the published version of the manuscript.

**Funding:** This research was funded by Russian Science Foundation, grant number 22-14-00026.

**Institutional Review Board Statement:** Not applicable.

**Data Availability Statement:** Not applicable.

**Acknowledgments:** We are grateful to O.G. Brekhov (Volgograd, Russia), D.A. Klyomin (Kazan, Russia), R.A. Kutushev (Nizhnekamsk, Russia), T.A. Chuzhekova (Odessa, Ukraine), A.V. Kovalev (Saint Petersburg, Russia), E.Yu. Rodionova (Riga) (Krasnodar, Russia), V.V. Anikin, (Saratov, Russia), K.V. Makarov, A.S. Prosvirov, I.A. Zabaluev (Moscow, Russia) and Yu. Volkova (Ulyanovsk, Russia) for material provided by them and to A.A. Prokin (Borok, Russia) for checking the identifications of some Dytiscidae and Hydrophilidae species.

**Conflicts of Interest:** The authors declare no conflict of interest.

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
