# Peer review of "Biodiversity of Coleoptera (Insecta) in the Middle and Lower Volga Regions (Russia)"

_diversity, doi:10.3390/d14121128_

Round 1

Reviewer 1 Report

The aim of the paper entitled “Biodiversity of Coleoptera (Insecta) in the Middle and Lower Volga regions (Russia)” is to describe the modern Coleoptera fauna in 9 regions of Russia on the basis of a recently published dataset, which contains data on 1469 species and subspecies of Coleoptera from 85 families (31433 samples and 9072 occurrences). The species of Coleoptera with range boundaries in the macroregion, as well as invasive Coleoptera species have been recorded.

The manuscript is clear, relevant for the field, scientifically sound and presented in a well-structured manner. The cited references are relevant and do not include an excessive number of self-citations. The paper is original (not published as far I know) and provide new data concerning the group of interest. There are many other beetles species will be found in the region (I suppose, the mentioned 1469 species are approximately 40-50% of possible fauna), but such work is important and the publication of the obtained results is useful and necessary. I think the paper is good and can be published (accepted for publication as it is).

Author Response

Dear reviewer,

We are grateful to you for the good analysis of our manuscript.

Reviewer 2 Report

The paper shortly describes the results of a study that provided information on the Coleoptera fauna of a region of Russia. However, it is a pity that the incredible amount of data was not more deeply analyzed.

There is here a list of species that reach their northernmost limits of distribution (better than northern borders, the same for easternmost, etc.). However, the authors should indicate the data from which the distribution of these species was taken (the Palaearctic catalogue gives only the country, so it does not give distribution with a suffcient detail). The simple list of species in random order is scarcely useful. First of all, the family should be indicated with an entry, and the species should be listed below each family. Secondarily, considering the large numberof species that have been collected or cited, it seems likely that there are more species that can reach their extremes of distribution, even though in many cases the details of their distribution may not be precisely known. The absence of any citation for the distribution of the various species renders difficult to understand the meaning of this part.

The large number of species would allow a much more complete analysis of the chorology, using also biogeographic categories. If such a study were made, a really important information of the biogeographic characterization of the fauna of the region could be achieved. As the authors themselves explain in the introduction, a camplete inventory of the fauna of a region is very important for a large amonut of studies. Since the authors give here a little information on the distribution of a few species, they should consider the hypothesis of supplying a much more complete analysis, including biogeography, percentages of the elements in the various biogeographic categories, and so on, so that the Coleoptera fauna of the region is truly described.

It is not clear why chapter 2, that are the resutls of the study, precedes chapter 3, Material and methods. It should be more appropriate to have Matherial and Methods as chapter 2, and the discussion of the results, Data Description, as chapter 3 - this in fact includes also a discussion on the distribution of some species.

Line 87: Replace "Such families were represented ..." with "Some families were represented..."

Author Response

The paper shortly describes the results of a study that provided information on the Coleoptera fauna of a region of Russia. However, it is a pity that the incredible amount of data was not more deeply analyzed.

There is here a list of species that reach their northernmost limits of distribution (better than northern borders, the same for easternmost, etc.). However, the authors should indicate the data from which the distribution of these species was taken (the Palaearctic catalogue gives only the country, so it does not give distribution with a suffcient detail). The simple list of species in random order is scarcely useful. First of all, the family should be indicated with an entry, and the species should be listed below each family. Secondarily, considering the large numberof species that have been collected or cited, it seems likely that there are more species that can reach their extremes of distribution, even though in many cases the details of their distribution may not be precisely known. The absence of any citation for the distribution of the various species renders difficult to understand the meaning of this part.

Answer: In our manuscript, we indicated the species that have the northern, eastern and western limits of their distribution within the macroregion. This information is given on the basis of the results of many years of research done by many authors. Our field observations are not related to the Palearctic Catalog, which gives only a general description of the distribution. In fact, these data are the result of many years of research by many authors of the manuscript.

The large number of species would allow a much more complete analysis of the chorology, using also biogeographic categories. If such a study were made, a really important information of the biogeographic characterization of the fauna of the region could be achieved. As the authors themselves explain in the introduction, a camplete inventory of the fauna of a region is very important for a large amonut of studies. Since the authors give here a little information on the distribution of a few species, they should consider the hypothesis of supplying a much more complete analysis, including biogeography, percentages of the elements in the various biogeographic categories, and so on, so that the Coleoptera fauna of the region is truly described.

Answer: The description of the dataset does not imply a significant analysis of the data obtained. That is, horology was not the purpose of the description in the manuscript. The manuscript summarizes the results of many years of research. But based on the compilation of the database, we now know where we have "white spots", what we need to pay attention to, where to spend more time researching in the future. It seems to us that future results in comparison with the data already obtained will allow us to give the information indicated by the reviewer. 

It is not clear why chapter 2, that are the resutls of the study, precedes chapter 3, Material and methods. It should be more appropriate to have Matherial and Methods as chapter 2, and the discussion of the results, Data Description, as chapter 3 - this in fact includes also a discussion on the distribution of some species.

Answer: The sections and subsections of the manuscript are made on the basis of the manuscript preparation form for this journal. We strictly adhered to these recommendations.

Line 87: Replace "Such families were represented ..." with "Some families were represented..."

Answer: corrected.

Reviewer 3 Report

The manuscript presents a brief description of a dataset. The manuscript organization is, however, poor and needs to be profoundly revised.

1. From Table 1, eventDate refers to only to materials from traps, but the database include also data collected by other forms of sampling (see lines 184-9). This point need revision.

2. Table 1. The expression “individuals represented present at the time of the occurrence” sounds weird to me. Do you mean “individuals found at the time of the observation”?

3. Lines 82-83. The authors say that “The total number of specimens in the dataset is 9072; the number of specimens represented is 31433”. I cannot understand the difference between “number of specimens in the dataset” and “number of specimens represented”.

4. Lines 91-93 are senseless. Perhaps the authors want to say “Many species have the northern limit of their distribution in the study area”. Or what else?

5. Lines 118-19 are obscure. Perhaps the authors want to say Valgus hemipterus (Linnaeus, 1758) and Notoxus binotatus (Gebler, 1829) found there their eastern and western limit, respectively”.

6. Line 137: What do you mean with “active”?

7. The Methods section should be placed before the Results section

8. Figure 1 must be redrawn. There is no need to have a double map for the same area. This confounds the reader. Only the part in colour is necessary, but add coordinates. Note that scale bar is left cut; correct.

9. The following references are inadequate for general statements and must be replaced with studies with broader scopes:3, 6, 7, 8, 11, 12, 13, 14, 15, 17, 19

Minor points

Table 1. Are you sure that Of requires the capital letter?

Lines 70 and 77 must be deleted.

Line 87: Such -> The following

121: The findings on invasive Coleoptera species are interesting -> Records of invasive species are particularly interesting.

162: elevated nature -> mountainous character

163: line 136 is grammatically senseless. Revise.

165-173: Provide references for this climatic description

195: modern data -> the most recent taxonomic arrangements

196: and also according to other publications -> and other publications

198: the reference book -> a reference book

Author Response

The manuscript presents a brief description of a dataset. The manuscript organization is, however, poor and needs to be profoundly revised.1.    From Table 1, eventDate refers to only to materials from traps, but the database include also data collected by other forms of sampling (see lines 184-9). This point need revision.

Answer: Collection date in all cases. Corrected

2.   Table 1. The expression “individuals represented present at the time of the occurrence” sounds weird to me. Do you mean “individuals found at the time of the observation”?

Answer: This is standard Darwin Core terminology: http://rs.tdwg.org/dwc/terms/version/individualCount-2021-07-15.htm “Individual” can be understood like “specimen”.

3. Lines 82-83. The authors say that “The total number of specimens in the dataset is 9072; the number of specimens represented is 31433”. I cannot understand the difference between “number of specimens in the dataset” and “number of specimens represented”.

Answer: 9072 – occurrences, 31433 - specimens (individuals). Corrected.

4. Lines 91-93 are senseless. Perhaps the authors want to say “Many species have the northern limit of their distribution in the study area”. Or what else?

Answer: Yes, this is a more concise expression of thought. Corrected.

5. Lines 118-19 are obscure. Perhaps the authors want to say Valgus hemipterus (Linnaeus, 1758) and Notoxus binotatus (Gebler, 1829) found there their eastern and western limit, respectively”.

Answer: You're right. That's what we meant.

6. Line 137: What do you mean with “active”?

Answer: We mean “The active expansion of area…”. Corrected

7. The Methods section should be placed before the Results section

Answer: The manuscript was prepared according to the template used in the journal. We strictly adhered to the template.

8. Figure 1 must be redrawn. There is no need to have a double map for the same area. This confounds the reader. Only the part in colour is necessary, but add coordinates. Note that scale bar is left cut; correct.

Answer: If the editor insists on redrawing the map, we will do it. One map is a simple diagram, while the other map gives an idea of the location of the macroregion of research and terrain. Both maps together better show the places of research. The scale is not cut off. It just starts with zero.

9. The following references are inadequate for general statements and must be replaced with studies with broader scopes:3, 6, 7, 8, 11, 12, 13, 14, 15, 17, 19

Answer: The following references indicate specific examples of a reduction in the number of species, a local reduction in insect populations, examples of changes in species ranges under the influence of anotropogenic causes. By this we show that changes on a local scale need to be taken into account and understood, since such local disturbances in populations can then occur in other parts of the range of a particular insect species. Nevertheless, we took into account the remark and replaced several references.

Minor points
Table 1. Are you sure that Of requires the capital letter?
Lines 70 and 77 must be deleted.

Answer: All column names are given according to the Darwin Core standard

Line 87: Such -> The following

Answer: Corrected.

121: The findings on invasive Coleoptera species are interesting -> Records of invasive species are particularly interesting.

Answer: Corrected.

162: elevated nature -> mountainous character

Answer: Corrected.

163: line 136 is grammatically senseless. Revise.

Answer: Corrected.

165-173: Provide references for this climatic description

Answer: Climatic description was taken from cited reference: 38. Milkov…

195: modern data -> the most recent taxonomic arrangements

Answer: Corrected.

196: and also according to other publications -> and other publications

Answer: Corrected.

198: the reference book -> a reference book

Answer: Corrected.

Round 2

Reviewer 2 Report

OK, The authors have explained their point of view regarding my comments.

Author Response

Thank for the review!

Reviewer 3 Report

The authors have not fully integrated my corrections.

Table 1. The expression “individuals represented present at the time of the occurrence” is wrong and different from that of http://rs.tdwg.org/dwc/terms/version/individualCount-2021-07-15.htm, which is

The number of individuals present at the time of the Occurrence.

YOU SHOULD DELETE REPRESENTED!

Line 121-122: As already indicate, this sentence should be corrected as follows: Valgus hemipterus (Linnaeus, 1758) and Notoxus binotatus (Gebler, 1829) have here their eastern and western limits, respectively”.

Line 140: The expression: “The active expansion of area of Harmonia axyridis” is very unclear. Replace with: “The current spreading of Harmonia axyridis out of its native range”

As already stated, Figure 1 must be redrawn. There is no need to have a double map for the same area. This confounds the reader. Only the part in colour is necessary, but add coordinates. They are essential! Note that scale bar is cut off (the 0 value is not completely displayed – pay attention!)

As already indicated, lines 165-6 are grammatically senseless: The mountainous character of the territory and the abundance of forests in many northern areas to a certain extent levels out the climatic differences. Please, revise.

References 3, 6, 7, 8, 11, 12, 13, 14, 15, 17, 19 are papers of very local interest. They should be replaced by papers of wider importance for an international audience. There are many review papers or even entire books that cover these issues that should be cited insteda of single case studies of only national interest.

Author Response

Table 1. The expression “individuals represented present at the time of the occurrence” is wrong and different from that of http://rs.tdwg.org/dwc/terms/version/individualCount-2021-07-15.htm, which is

The number of individuals present at the time of the Occurrence.

YOU SHOULD DELETE REPRESENTED!

Answer: corrected.

Line 121-122: As already indicate, this sentence should be corrected as follows: Valgus hemipterus (Linnaeus, 1758) and Notoxus binotatus (Gebler, 1829) have here their eastern and western limits, respectively”.

Answer: corrected.

Line 140: The expression: “The active expansion of area of Harmonia axyridis” is very unclear. Replace with: “The current spreading of Harmonia axyridis out of its native range”

Answer: corrected.

As already stated, Figure 1 must be redrawn. There is no need to have a double map for the same area. This confounds the reader. Only the part in colour is necessary, but add coordinates. They are essential! Note that scale bar is cut off (the 0 value is not completely displayed – pay attention!)

Answer: A new map has been made.

As already indicated, lines 165-6 are grammatically senseless: The mountainous character of the territory and the abundance of forests in many northern areas to a certain extent levels out the climatic differences. Please, revise.

Answer: Sentence "Nevertheless, climatic differences in the macro-region are quite pronounced has been removed"

References 3, 6, 7, 8, 11, 12, 13, 14, 15, 17, 19 are papers of very local interest. They should be replaced by papers of wider importance for an international audience. There are many review papers or even entire books that cover these issues that should be cited insteda of single case studies of only national interest.

Answer:  We have replaced references 6, 7, 8, 12, 13, 19 with papers of wider importance. 

Round 3

Reviewer 3 Report

1) The problematic sentence is not that you have removed, but this one: " The elevated nature of the territory and the abundance of forests in many northern  areas to a certain extent levels out the climatic differences". The sentence "Climatic differences in the macro-region are quite pronounced".

2) The sentences "There are the northern borders of distribution of many Coleoptera species in the macro-region. Due to the considerable latitudinal extent of the studied macro-region, a  large number of the following species have their northern borders of distribution" are unclear. Do you mean that only a part of the "following species" have here their northern borders of distribution? If so, delete the species that do not have there their northern borders of distribution I the region and cite only those that have their northern borders of distribution. And change as follows: "There are the northern borders of distribution of many Coleoptera species in the macro-region. Due to the considerable latitudinal extent of the studied macro-region, a  large number of the following species have their northern borders of distribution" -> "Many Coleoptera species found in the macro-region their northern borders of distribution".

Author Response

1) The problematic sentence is not that you have removed, but this one: " The elevated nature of the territory and the abundance of forests in many northern  areas to a certain extent levels out the climatic differences". The sentence "Climatic differences in the macro-region are quite pronounced".

Answer:  The sentence "The elevated nature of the territory and the abundance of forests in many northern  areas to a certain extent levels out the climatic differences" was replaced by "Climatic differences in the macro-region are quite pronounced"

2) The sentences "There are the northern borders of distribution of many Coleoptera species in the macro-region. Due to the considerable latitudinal extent of the studied macro-region, a  large number of the following species have their northern borders of distribution" are unclear. Do you mean that only a part of the "following species" have here their northern borders of distribution? If so, delete the species that do not have there their northern borders of distribution I the region and cite only those that have their northern borders of distribution. And change as follows: "There are the northern borders of distribution of many Coleoptera species in the macro-region. Due to the considerable latitudinal extent of the studied macro-region, a  large number of the following species have their northern borders of distribution" -> "Many Coleoptera species found in the macro-region their northern borders of distribution".

Answer: We have restructured the sentence as follows: "There are the borders of distribution of many Coleoptera species in the macro-region. Due to the considerable latitudinal extent of the studied macro-region, a large number of the species have their northern borders of distribution:..."